# Effect of CO_2_ Content in Air on the Activity of Carbonic Anhydrases in Cytoplasm, Chloroplasts, and Mitochondria and the Expression Level of Carbonic Anhydrase Genes of the α- and β-Families in *Arabidopsis thaliana* Leaves

**DOI:** 10.3390/plants11162113

**Published:** 2022-08-14

**Authors:** Natalia N. Rudenko, Lyudmila K. Ignatova, Ilya A. Naydov, Natalia S. Novichkova, Boris N. Ivanov

**Affiliations:** Institute of Basic Biological Problems of the Russian Academy of Sciences, Federal Research Center “Pushchino Scientific Center for Biological Research of the Russian Academy of Sciences”, 142290 Pushchino, Moscow Region, Russia

**Keywords:** carbonic anhydrase, CA activity, photosynthesis, carbon assimilation, *Arabidopsis*, gene expression, cytoplasm, chloroplasts, thylakoids

## Abstract

The carbonic anhydrase (CA) activities of the preparations of cytoplasm, mitochondria, chloroplast stroma, and chloroplast thylakoids, as well as the expression levels of genes encoding αCA1, αCA2, αCA4, βCA1, βCA2, βCA3, βCA4, βCA5, and βCA6, were measured in the leaves of *Arabidopsis thaliana* plants, acclimated to different CO_2_ content in the air: low (150 ppm, lCO_2_), normal (450 ppm, nCO_2_), and high (1200 ppm, hCO_2_). To evaluate the photosynthetic apparatus operation, the carbon assimilation and chlorophyll *a* fluorescence were measured under the same conditions. It was found that the CA activities of the preparations of cytoplasm, chloroplast stroma, and chloroplast thylakoids measured after two weeks of acclimation were higher, the lower CO_2_ concentration in the air. That was preceded by an increase in the expression levels of genes encoding the cytoplasmic form of βCA1, and other cytoplasmic CAs, βCA2, βCA3, and βCA4, as well as of the chloroplast CAs, βCA5, and the stromal forms of βCA1 in a short-term range 1–2 days after the beginning of the acclimation. The dependence on the CO_2_ content in the air was most noticeable for the CA activity of the preparations of the stroma; it was two orders higher in lCO_2_ plants than in hCO_2_ plants. The CA activity of thylakoid membranes from lCO_2_ plants was higher than that in nCO_2_ and hCO_2_ plants; however, in these plants, a significant increase in the expression levels of the genes encoding αCA2 and αCA4 located in thylakoid membranes was not observed. The CA activity of mitochondria and the expression level of the mitochondrial βCA6 gene did not depend on the content of carbon dioxide. Taken together, the data implied that in the higher plants, the supply of inorganic carbon to carboxylation sites is carried out with the cooperative functioning of CAs located in the cytoplasm and CAs located in the chloroplasts.

## 1. Introduction

In all living organisms, cells have several buffer systems, including those based on forms of Ci, CO_2_, and HCO_3_^−^. This is, apparently, one of the main reasons why the enzyme carbonic anhydrase (CA), which accelerates the interconversion of CO_2_ and bicarbonate with the release and consumption of a proton, emerged at least eight times in the course of evolution in different groups of living organisms.

In autotrophs, which include inorganic carbon (Ci) into organic compounds, the interconversion of Ci forms is of particular importance. On the way from the medium to the carboxylation centers, where the fixation of inorganic carbon into organic compounds takes place, this interconversion must repeatedly occur to overcome the membranes and aqueous phases of cells. The presence of CAs is required to accelerate the feeding of the necessary Ci form and to ensure a high rate of transport. In unicellular photoautotrophs, bacteria and green microalgae, CAs function in the system of Ci concentration in the cells as compared with that of the environment. In all plants, CAs are needed to maintain the required local concentration of a particular form of Ci near the carboxylation sites. The substrate for ribulose-1,5-bisphosphate carboxylase-oxygenase (Rubisco) in C3 plants and in bundle sheath cells of C4 plants is CO_2_, and in the mesophyll cells of C4 plants, a substrate for phosphoenolpyruvate carboxylase (PEP carboxylase) is bicarbonate. In higher C3 plants, despite the attempts to study the mechanisms of CO_2_ entry into cells to the sites of their fixation in metabolic pathways and, first of all, to Rubisco, the proven statements on how CAs participate in the conversion of Ci forms to achieve the physiologically necessary result are almost absent.

In cells of angiosperms, as well as of most algae, there are representatives of three families, α, β, and γ, which are soluble or membrane-bound, and have a wide range of molecular masses and pH optima. In the genomes of higher C3 plants, the number of genes encoding CAs ranges from four in *Spirodela polyrhiza* to seventy-nine in *Triticum aestivum*.

CAs are present in all cell structures. In plasma, the membrane of Arabidopsis leaves βCA4 was found [1]. The other form of this enzyme was found in cytoplasm [2], with three other CAs, βCA2, and βCA3 [1], and βCA1.4, which has been recently determined as the cytoplasmic form of βCA1 [3]. In mitochondria, βCA6 was found in the mitochondrial matrix [1] and all five γCAs were identified as a part of the CA-domain attached to the inner surface of the mitochondrial membrane [4].

In chloroplasts, there are several CAs of β- and α-families. Two forms of βCA1, βCA1.1 and βCA1.2 [3], are situated in chloroplast stroma as well as αCA1 [5]. The presence of βCA1.3 was detected in the chloroplast envelope [3]. βCA5 is one more chloroplast CAs [1] with an unknown precise position in this organoid. The thylakoid lumen of pea and Arabidopsis chloroplasts possess CA activity [6,7]. Based on a number of properties, this CA was identified as βCA, but not βCA1, and it was assumed that this CA was βCA5 [8,9]. Proteomic analysis of proteins in thylakoid membranes from Arabidopsis revealed the product of the *At4g20990* gene encoding αCA4 [10]. Ignatova et al. [11] showed that this CA is situated in the thylakoid membrane close to PSII complexes. In [12], the evidence was presented that the position of αCA2 is also in the thylakoid membranes. The presence of αCA5 in the stromal parts of thylakoid membranes of Arabidopsis was revealed using mass-spectrometry analysis [13].

The data of the literature indicate that CAs function mutually with other CAs. Wang et al. [14] have shown the involvement of the plasma membrane form of βCA4 in complex with aquaporin PIP2;1 in the enhancement of CO_2_ transport in Arabidopsis cells. Hu et al. [15], using Δβ-ca1ca4 double mutants, showed that βCA4 regulates the stomatal conductance of leaves in the presence of carbon dioxide together with chloroplast stromal βCA1. Under low CO_2_ conditions, the growth of the mutants with knocked-out genes of two cytoplasmic CAs, βCA2, and βCA4, was significantly suppressed [2], which means that at least βCA1, βCA2, and βCA4 participate in the process of carbon dioxide entry into higher plants cells.

The main hypotheses on the physiological role of chloroplast stromal CAs, taking into account the changing levels of substrates and products catalyzed by the CA reaction, were formulated by Coleman [16]: (1) facilitating Ci fluxes in the stroma to the site of Rubisco-mediated carboxylation; (2) the catalysis of bicarbonate dehydration to CO_2_ in the alkaline stroma to supply the latter for Rubisco; and (3) the acceleration of pH changes in biological buffering systems. Despite copious evidence of a spatial and functional connection of stromal βCA (βCA1) and Rubisco [17,18,19,20,21], no definite effects from the inhibition of this CA synthesis on photosynthesis have been shown. Price et al. [22], in transgenic plants of *Nicotiana tabacum* with 10% or less activity of soluble βCA, have not found significant differences from in a number of parameters characterizing photosynthesis compared to WT plants. However, the carbon isotopic composition of the leaf dry matter was changed in these mutants. One of the promising ideas in solving the problem of the conversion of bicarbonate to CO_2_ and the supply of the latter to Rubisco is the assumption that thylakoid CAs are involved in this process, additionally to the stromal CAs, αCA1, and/or βCA1 [23]. Most likely, this function can be carried out by αCA5, since this CA is situated on the stromal side of stromal thylakoid membranes, where the part of Rubisco associated with the thylakoid membrane is located [24,25].

The presence of Ci forms is important not only for the dark phase of photosynthesis but also for the optimal functioning of the electron transport chain. Bicarbonate is required for the electron transport on the acceptor side of PSII [26]. HCO_3_^−^ interaction with non-heme iron ensures rapid electron transfer from Q_A_ to Q_B_ [27]. Bicarbonate is also required on the donor side of PSII, and it was suggested that CA located there could be involved in HCO_3_^−^ delivery [27,28]. Studies on the functioning of αCA4 in Arabidopsis thylakoid implied that this CA is located close to light-harvesting PSII antennae [10,11,29]. The assumption about the role of αCA4 is that it directs the protons that arise from CO_2_ hydration to the proteins (PsbS and/or violaxanthin de-epoxidase), which initiate the non-photochemical quenching of leaf chlorophyll *a* fluorescence (NPQ) [30,31,32]. These processes protect PSII from photoinhibition. Because of the CO_2_ hydration bicarbonate molecules released in this reaction, they can bind with protons, which produce during water oxidation in the water-oxidizing PSII complex [33].

In recent years, more data on the involvement of CAs in higher plant cells for protection against stresses, in particular in transmitting a response signal to a negative environmental impact, have begun to appear [34,35,36,37]. Dabrowska-Bronk et al. [38] have shown that Arabidopsis plants with single mutations in all six βCAs genes had a weakened stress tolerance under high light conditions. At the same time, these authors emphasize that the regulation of βCAs’ gene expression and the enzymatic activities of CAs are important for optimal plant growth and photosynthesis.

An understanding of the physiological roles of CAs in higher plants is still something elusive. Studies using mutants with the suppressed synthesis of CA genes usually turn out to be of little information, and we assume that the reason for this may be that CAs in plants cell function together, and some can replace the weakened function of one of them. The main aim of the present work was to determine the changes in the activity of CAs in different compartments of plant cells in parallel with the expression level of the genes encoding these CAs by exposing Arabidopsis plants to altered carbon dioxide content in the air, which is the main nutrition source for plants and the main substrate of CA reaction.

## 2. Results

### 2.1. Photosynthetic and Stress State Characteristics of Arabidopsis Plants during Acclimation to Changed CO_2_ Conditions

The acclimation of Arabidopsis plants grown under normal CO_2_ levels in the air (450 ppm, nCO_2_) to changed CO_2_ levels in the air, low (150 ppm, lCO_2_) and high (1200 ppm, hCO_2_), was measured as changes of the induction of the chlorophyll *a* fluorescence before and 3, 9, and 16 days after exposure to changed CO_2_ content. Plants grown at nCO_2_ were used as a control for lCO_2_ and hCO_2_ plants of the same age.

In lCO_2_- and hCO_2_-acclimated plants versus nCO_2_ plants, all parameters of OJIP kinetics commonly used for stress evaluation were changed. The photosynthetic performance index of PSII (PI_abs_), which is estimated to absorption energy per the photochemically active reaction center of PSII, as well as the maximum quantum yield of PSII (Fv/Fm), decreased in the lCO_2_ plants (Figure 1A,B) after three days, whereas in hCO_2_ plants, a decrease in these parameters was observed only after 9 and 16 days, respectively. These changes were in parallel with an increase in the dissipation of light energy per the active reaction center (DIo/RC), i.e., a parameter which characterizes the activation of the defense mechanisms of the photosynthetic apparatus in response to stress, also three days after exposure to lCO_2_ and 16 days after the exposure to hCO_2_ (Figure 1C). The effect of the carbon dioxide level on the performance index of the total photosynthetic electron chain (PI_total_) was weaker than on PI_abs_, Fv/Fm, and DIo/RC (Figure 1D). A decrease in the PI_total_ in hCO_2_ and lCO_2_ plants compared to nCO_2_ plants has appeared only by the ninth day.

After 16 days of acclimation to low, normal, and high CO_2_ content in the air, we have measured the expression level (expression intensity) of the genes, which are the markers of the induction of stress transcriptional cascades activated by the molecules of phytohormones: abscisic acid (ABA), jasmonic acid (JA) and salicylic acid (SA) (Figure 2). ABA and JA are synergists and both of them are antagonists of SA [39]. *At1g29395* and *At1g52890* are both ABA-induced genes. *At1g29395* encodes the integral membrane protein Cold Regulated 414 Thylakoid Membrane 1 (COR414-TM1), located in the inner envelope of chloroplasts, which provides tolerance to cold and water deprivation. *At1g52890* encodes NAC Domain Containing Protein19, one of the main transcription factors in plant abiotic stress responses. *At1g17420* (*lox3*) and *At5g42650* (*aos*) are induced by JA. *At1g74710* encodes a lipoxygenase, which catalyzes the oxygenation of fatty acids. *At5g42650* encodes the allene oxide synthase that catalyzes the dehydration of the hydroperoxide to oxide in the JA biosynthetic pathway. *At1g74710* (*icsi*), *At3g52430* (*pad*), and *At1g64280* (*npr1*) are SA-induced genes. *At1g74710* encodes a protein with isochorismate synthase activity, which is important for SA accumulation. *At3g52430* encodes a lipase-like gene that is important for salicylic acid signaling. The nonexpressor of Pathogenesis-Related (NPR1) is a key regulator of the SA-mediated systemic acquired resistance (SAR) pathway.

In lCO_2_ plants, the contents of the transcripts of ABA- and JA-inducible genes *At1g29395*, *At1g17420* (*lox3*), and *At5g42650* (*aos*) were 16, 10, and 3 times higher than in nCO_2_ plants, respectively (Figure 2A,B). The expression levels of SA-induced genes *At3g52430* (*pad*) and *At1g64280* (*npr1*) were 7 and 15 times lower in lCO_2_ plants than in nCO_2_ plants (Figure 2C). 

In hCO_2_ plants, the response of stress marker genes was less expressed. Only a four- to five-fold increase in the content of the transcripts of the SA-induced *At1g74710* (*icsi*) gene versus nCO_2_ plants was observed (Figure 2C). The other analyzed stress-marker genes had a tendency to be 10–50% higher in hCO_2_ plants than in nCO_2_ plants (Figure 2A–C).

### 2.2. CO_2_ Assimilation and Carbon Levels in Leaves of Arabidopsis Plants Acclimated to Changed CO_2_ Content in the Air

The CO_2_ assimilation rate was measured in Arabidopsis plant leaves after 9 (Figure 3A) and 14–16 (Figure 3B) days of exposure to lCO_2_ and hCO_2_ with nCO_2_ plants of the same age as a control. For this purpose, the measurement chamber of the LI-6800 Portable Photosynthesis System using the dynamic assimilation technique at CO_2_ concentrations varying from 0 to 1200 ppm has been used. By the ninth day of acclimation, the difference in CO_2_ assimilation rate between lCO_2_, nCO_2_, and hCO_2_ plants was imperceptible (Figure 3A). This difference became noticeable after about two weeks after the exposure to the changed CO_2_ level (Figure 3B). In lCO_2_ plants, the CO_2_ assimilation rate was lower than in nCO_2_ plants. This difference was the highest in atmospheric CO_2_ content in the measurement chamber.

The lowest CO_2_ assimilation rate has been detected in the hCO_2_ plants (Figure 3B), which was the most expressed at 1200 ppm in the measurement chamber. Photosynthetic down-regulation in the hCO_2_ plants (Figure 1B), which has also been observed by Zheng et al. [40] at elevated CO_2_ levels, could be the reason for the relatively low CO_2_ assimilation rate in the hCO_2_ plants against the nCO_2_ and even the lCO_2_ ones.

The carbon fixation products, starch and soluble carbohydrates, were determined in the Arabidopsis plants’ leaves after 16 days of acclimation to the changed CO_2_ level in the air. It was the lowest in the lCO_2_ plants’ leaves (Table 1). The starch content in these plants was 70–80% lower, and the content of soluble carbohydrates was 20–30% lower than in the nCO_2_ plants. In the hCO_2_ plants, the content of starch was also lower than in the nCO_2_ plants by about 30–40%, which correlates with the lower CO_2_ assimilation rate in these plants (Figure 3B). However, the content of soluble carbohydrates was slightly higher in the hCO_2_ plants than in the nCO_2_ ones.

Thus, the presented results demonstrate that Arabidopsis plants grown at short day photoperiods and low photosynthetically active radiation (see Materials and Methods) require at least two weeks of acclimation to the changed CO_2_ content in the air, both low and high.

### 2.3. CA Activity in Preparations of Cytoplasm, Mitochondria, and Chloroplasts Isolated from Leaves of Plants Acclimated to Low, Normal, and High CO_2_ Level in the Air

We have found that CA activity in preparations of cytoplasm, where βCA4, βCA2, βCA1.4, and βCA3 are located, was about 30% higher in lCO_2_ plants than in nCO_2_ and hCO_2_ ones (Figure 4A). The CA activity of mitochondria preparations incubated with 1% Triton X-100 was independent of the CO_2_ level in the air (Figure 4B).

The CA activities of preparations isolated from chloroplasts were the most sensitive to the CO_2_ levels in the air (Figure 4C–E). We measured the CA activity of the stroma and thylakoids separately. The CA activity of the preparations of chloroplast stroma, where βCA1.1, βCA1.2, and αCA1 are located, increased three-fold in the lCO_2_ plants compared to the nCO_2_ plants. In the hCO_2_ plants, the CA activity in these preparations was reduced to 3% of that in the nCO_2_ plants. Thus, the activity of the CAs in the stroma of the lCO_2_ plants was two orders higher than of those in the stroma of the hCO_2_ plants (Figure 4C).

The thylakoids during the isolation process were thoroughly washed from extrathylakoid CAs according to [7,41,42]. The CA activity of the thylakoids was determined after incubation with the detergent Triton X-100. The addition of Triton X-100 at a Triton/Chl ratio of 0.3 (Thyl.0.3) exhibits the maximum activity of the CA located in the stromal parts of the thylakoid membranes, i.e., close to PSI [6,42,43,44]. We have recently identified it as αCA5 [13]. The CA activity of Thyl.0.3 increased by about six times after acclimation to 150 ppm of CO_2_ and by 2.5 times after acclimation to 1200 ppm of CO_2_, respectively (Figure 4D), if compared with nCO_2_ plants.

The CA activity of CAs located in the granal thylakoid membranes, i.e., close to PSII, shows its maximum after the incubation of the thylakoids with Triton X-100 at a Triton/Chl ratio of 1.0 (Thyl.1.0) [6,42,43,44]. This CA activity is apparently determined due to the presence of αCA4 [10,11,30,31,32] and αCA2 [11]. The CA activity of Thyl.1.0 was twice higher in the lCO_2_ plants than in the nCO_2_ plants (Figure 4E). In the hCO_2_ plants, the CA activity of Thyl.1.0 was significantly, about 97%, lower than in the nCO_2_ plants, i.e., the activity of the CAs from the granal thylakoid membranes of the plants acclimated to the lCO_2_ level in the air was 30 times higher than of those from plants acclimated to the high CO_2_ level.

### 2.4. The Effect of Acclimation of Adult nCO_2_ Plants to Changed CO_2_ Content in the Air on the Expression Level of the Genes Encoding CAs of α- and β-Families

We have measured the expression level, i.e., the content of transcripts, of the genes encoding αCA1, αCA2, αCA4, βCA1, βCA2, βCA3, βCA4, βCA5, and βCA6 in Arabidopsis leaves of different ages grown at nCO_2_ (Table 2). The content of the transcripts of the gene encoding thylakoid αCA5, located in stromal thylakoid membranes, i.e., close to PSI, has not been measured, since it is too low to be detected by the Real-Time RT PCR [13]. There are four alternative splicing forms of the βCA1 gene, with two pairs with the same sequences on 3′ ends: *βca1.1 + βca1.2* and *βca1.3 + βca1.4*. We have determined the correspondent pairs together and denoted them as *βca1.1+1.2* and *βca1.3+1.4*. βCA1.1 and βCA1.2 are situated in the chloroplasts [3], and the *βca1.1+1.2* transcripts were related to the group of chloroplast CAs. The other form of βCA1, βCA1.3, was determined in the chloroplast envelope, whereas βCA1.4 was detected in the cytoplasm with a much stronger GFP fluorescence signal than for βCA1.3 [3]. Thereafter, we related the *βca1.3+1.4* transcripts to the group of extrachloroplast CAs. Alternative splicing forms of the βCA4 gene, which encodes βCA4.2 located in the cytosol [2] and βCA4.1 located in the plasma membrane [2], also possess the same sequences on 3′ ends of exons, and the correspondent transcripts could not be defined separately.

The intensity of the expression of the CA genes in control plants under the constant CO_2_ level at the normal growth conditions, i.e., at atmospheric CO_2_ content, was the highest in the leaves of young, 26 days-old plants for most CA genes (Table 2) and gradually decreased with age. In the nCO_2_ plants, this expression level was rather stable from 44 to 52 days of age (Figure 5 and Figure 6, white columns) and decreased for all analyzed genes by the age of about 59–60 days. In addition, in 59–60 days (two months) old plants, the effects of CO_2_ on the intensity of the CAs genes expression, are absent (Figure 5 and Figure 6).

Figure 5 shows the content of the same transcripts (Table 2) in the leaves of adult plants during the acclimation to different CO_2_ content in the air. In two-month-old plants, the intensity of the expression of βCA4 did not show a significant change under lCO_2_ (Figure 5A). However, after 2 days of introduction to lCO_2_, it was about 30% higher than in the nCO_2_ plants of the same age. Its expression was not significantly changed in the lCO_2_ plants during CO_2_ acclimation.

Until recently, βCA2 was considered as the main CA in the cytoplasm of higher plants [45,46] due to its abundance [1,47] and high expression level [48,49]. The data in Table 2 also show that the expression level of *βca2* was one of the highest. However, the contents of the *βca4* and *βca1.3+1.4* transcripts are also high. The expression level of the genes encoding cytoplasmic Cas *βca2*, *βca1.3+1.4*, and *βca3* increased by two times on the second day under lCO_2_ (Figure 5B,C) versus nCO_2_. During further acclimation to lCO_2_, the expression level of all four genes of the cytoplasmic Cas decreased, and by the ninth day, became lower than in nCO_2_ (Figure 5A–D).

The significant effect of hCO_2_ has been observed only for the expression of *βca2*. On the first day of exposure to hCO_2_, it decreased to 12% from that in nCO_2_ and increased to 212% by the third day of acclimation (Figure 5B).

The expression level of the mitochondrial βCA6 was independent from CO_2_ concentration in the air during all the acclimation (Figure 5E). These data correspond with the constant CA activity in the preparations of mitochondria at any CO_2_ content in the air (Figure 4B).

The intensity of the expression of the genes encoding chloroplast CAs was fluctuating during acclimation (Figure 6). However, in lCO_2_ conditions versus nCO_2_, the reliable increase in the content of the *βca1.1+1.2* transcripts (Figure 6A) and the *βca5* transcripts (Figure 6E), eight and three times, respectively, have been observed. During further acclimation to lCO_2_, the expression levels of *βca1.1+1.2* and *βca5* decreased and became lower by the ninth day than in control plants, i.e., changed in the same way as the expression intensity of the genes of cytoplasmic CAs (Figure 5A–D). The expression level of the genes, encoding stromal αCA1 and thylakoid CAs αCA2 and αCA4 were not significantly dependent on CO_2_ content in the air (Figure 6B–D).

Thus, the observed high-CA activity of cytoplasmic and chloroplast preparations in two-month-old lCO_2_ plants if compared to nCO_2_ plants and, to an even greater extent, to hCO_2_ (Figure 4), was preceded by an increase in the expression levels of the genes encoding chloroplast βCA1 and βCA5 and cytoplasmic βCA2, βCA3, and βCA4 (Figure 5 and Figure 6) in a short-term range 1–2 days after the beginning of the acclimation.

### 2.5. The effect of Acclimation of Young Plants to low CO_2_ Content in the Air on the Expression Level of the Genes Encoding CAs of α- and β-Families

A low concentration of carbon dioxide in the air is a significant stress factor for plants (Figure 1 and Figure 2), leading to a considerable increase in the CA activity of most of the studied fractions (Figure 4), with a much less pronounced increase in gene expression intensity (Figure 5 and Figure 6). Therefore, the additional studies on changes in the levels of CA gene expression in Arabidopsis plants younger than two months of age after acclimation to lCO_2_ are noted. In 26 days-age plants, a decrease in the CO_2_ content for two weeks led to an increase in the expression levels of most genes of the cytoplasmic and chloroplast CAs (Figure 7A). This expression level was 1.5–2 times higher for *αca2* and *βca5*, 3–4 times higher for *βca2* and *βca3* genes, and 7–9 times higher for *βca1.1+1.2* and *βca1.3+1.4*. The expression level of only *αca4* was about 60% lower in lCO_2_ plants than in nCO_2_ plants.

In 50 days-age plants, 16 days of acclimation to lCO_2_ led to an increase in the intensity of the expression of other CA genes to a much lesser extent than in 26 days-age plants. The expression level of *βca4* and *αca1* increased by two times; the *αca4* and *βca5* gene expressions were about three times higher (Figure 7B). The expression levels of cytoplasmic *βca2*, *βca3* and *βca1.3+1.4* were about two to three times lower in lCO_2_ plants than in nCO_2_ plants.

## 3. Discussion

In the present study, we have determined the changes in CA activities not in the total leave extract from C3 higher plant *A. thaliana* in response to CO_2_ content in the air, as in previous studies [50,51], but in different cell fractions isolated from leaves of C3 higher plant *A. thaliana* acclimated to the low and high CO_2_ content in the air. The changes in the contents of transcripts of the correspondent CA genes during acclimation were analyzed in order to evaluate the response of their expression during such acclimations.

We have studied the above-listed changes when the negative consequences of acclimations had already taken place, manifesting primarily in a decrease in the maximum quantum yield of PSII (Fv/Fm) and performance indexes parameters (PI_total_ and P_abs_). In the plants exposed to lCO_2_, the changes in these photosynthetic parameters started in three days and were expectedly lower than in nCO_2_ plants (Figure 1). However, the decline in the carbon dioxide assimilation rate in these plants, as well as in the content of starch and soluble carbohydrates (Table 1) and the lCO_2_ plant size (not shown) has been observed not earlier than about two weeks after exposure to lCO_2_ (Figure 3B). These data are easily explained in terms of the shortage of the CO_2_ content, i.e., of the basic material for photosynthesis. At the same time, the parameters of photosynthesis in plants exposed to hCO_2_ also decreased versus those in nCO_2_ ones, although later than in lCO_2_ plants, only on the 9^th^ day for PI_total_ and PI_abs_ and on the 16^th^ day for Fv/Fm (Figure 1A,B,D). Despite the larger size of the hCO_2_ plants, the CO_2_ assimilation rate in their leaves and the content of starch (mg/g of fresh weight) were lower than in the nCO_2_ plants (Figure 3B, Table 1). Photosynthetic down-regulation in the hCO_2_ plants (Figure 1B), which has also been observed by Zheng et al. [40] at elevated CO_2_ levels, could be the reason for the relatively low CO_2_ assimilation rate in hCO_2_ plants against nCO_2_ and even lCO_2_ ones. This can be explained by the known effects of the changes in leaf configuration, i.e., the reduction in stomatal apertures and mesophyll tissue size at an elevated level of atmospheric CO_2_ [40,52]. Another reason for the decrease in the assimilation rate may be the acidification of the stroma as a result of indirect proton transfer because of the absorption of excessive CO_2_ [53]. Probably, in the conditions of the changed carbon dioxide level, some compensatory mechanisms in the total photosynthetic electron chain take place. Thus, in Arabidopsis plants, the maximum stress effect from the changed CO_2_ level developed on the 16^th^ day.

One of the questions raised by investigations of the effect of CO_2_ content changes is if a decrease in CO_2_ content increases stress, or if the stress weakens the plants’ response to changes in the CO_2_ level [48]. Our results imply the first conclusion. The dissipation of light energy per active reaction center (DI_0_/RC) in both lCO_2_ and hCO_2_ plants was higher than in nCO_2_ ones (Figure 1C). Changes in the expression of stress-induced genes showed that both lCO_2_ and hCO_2_ conditions are stressful for plants, wherein, in lCO_2_ plants and in hCO_2_ plants, apparently, different stress responses are induced. In lCO_2_ plants, the expression levels of the ABA- and JA-induced genes were higher than in nCO_2_ and hCO_2_ plants (Figure 2A,B), while the expression levels of the marker genes of SA pathways were lower. It is important that the expression of the *npr1* gene, which is the key regulator of SA-mediated SAR, was about seven times lower in lCO_2_ plants than in nCO_2_ and hCO_2_ ones (Figure 2C). Since it is known that ABA and JA are synergists and both of them are antagonists of SA [39], these data mean that in plants at lCO_2_, the ABA- and JA-induced SAR pathways are activated, while the SA-mediated pathway is suppressed.

Our data indicate that for Arabidopsis plants, not only the conditions of low levels but also the conditions of high levels of carbon dioxide in the air, are stressful, although to a lesser extent. This is demonstrated both by the intensity of expression of stress genes and by photosynthetic parameters (Figure 1 and Figure 2). The response of stress marker genes to the acclimation to hCO_2_ was less expressed than in lCO_2_ plants; for most genes analyzed, there was a small, only 10–50% increase in their expression in the hCO_2_ plants versus the nCO_2_ plants (Figure 2A–C).

The lipophilic CO_2_ molecule should easily diffuse across lipid membranes [54,55]. However, this diffusion through cell and organelle membranes is slowed down by a series of resistances due to a high content of protein and sterol molecules in these membranes [55,56]. The diffusion to the site of carboxylation can also be slowed down due to diffusion resistance in the aqueous phases of the cytoplasm and chloroplast stroma [57]. It has long been hypothesized that the CAs of the plasma membrane, cytoplasm, mitochondria, and chloroplasts are involved in Ci transport, in particular, in the supply of CO_2_ to carboxylation centers [8,16,47].

The CA activity of mitochondria is determined by the presence of the complex of γCAs-subunits attached to the inner mitochondrial membrane [58] and of the presence of βCA6 in the matrix [1]. Soto et al. [4] showed that the expression intensity of the encoding γCAs genes decreased under conditions of high CO_2_ content, and Fabre et al. [1], using semi-quantitative PCR, demonstrated a higher intensity of the band of PCR products in plants grown at high CO_2_ levels versus normal CO_2_. In our experiments, the CA activity of the mitochondria preparations incubated with 1% Triton X-100 and the contents of the *βca6* transcripts were independent of the CO_2_ level in the air (Figure 4B, Figure 5E and Figure 7A). One of the earlier hypotheses [59] about the role of mitochondrial CAs assumed their participation in Ci supply to chloroplasts under conditions of low CO_2_ in the apoplast, for example, under conditions leading to stomatal closure. Our data do not support these assumptions. Participation in the processes of dark respiration in mitochondria seems to be more probable. This is indicated by a significant increase in the intensity of *βca6* gene expression after 48 h in the dark [48].

The CA activities of the cytoplasm and all analyzed fractions of chloroplasts, i.e., preparations of stroma and thylakoids, were higher in 1CO_2_ plants, than in nCO_2_ plants (Figure 4A,C–E). At that, the CA activity of the preparations of the stroma and Thyl.1.0 (CA activity of granal thylakoid membranes) in 1CO_2_ plants turned out significantly higher than in hCO_2_ plants. These data imply that CAs located in the cytoplasm of photosynthesizing cells and CAs located in the chloroplasts, both in stroma and in thylakoids, participate in Ci supply to carboxylation sites in higher plants. Participation in the conversion of bicarbonate into CO_2_ to provide it to Rubisco in the stroma by several soluble CAs in parallel with thylakoid CAs seems to be even necessary. This would ensure the optimal rate of CO_2_ supply to the carboxylation centers, which is the most important physiological process for plants. At that, the cooperative participation of stromal and thylakoid CAs in this process would improve plants’ ability to adapt to changing environmental conditions. It seems most likely that of all the thylakoid CAs, it is αCA5, located on the stromal side of the stromal thylakoid membranes, which can be involved in this process [13]. The CA activity of the stromal thylakoids was the highest in lCO_2_ (Figure 4D). Herein, the activity of the stromal thylakoids was 2.5 times higher in hCO_2_ plants versus nCO_2_ plants. Under hCO_2_, the activity of the stromal thylakoids was even higher than the CA activity of the preparations of stroma (Figure 4C). These data imply that under high CO_2_, the function of CO_2_ supply to Rubisco is carried out, to a greater extent, by αCA5 than by the stromal CAs.

Under the lCO_2_ level, i.e., in conditions of a deficiency of a Calvin–Benson cycle substrate, the value of NPQ increases. This is confirmed by the increment of the DI_0_/RC parameter in lCO_2_ plants versus nCO_2_ ones (Figure 1B). The CA activity of the granal thylakoids under lCO_2_ is more likely determined by the need for the participation of αCA4 located here [10,11] in the development of NPQ [30,31,32] than by the involvement of granal thylakoid CAs in Ci transport.

The expression level of CA-encoding genes in higher plants is daytime-dependent. That was shown for *A. thaliana* [48] and CAM plants, *Sedum album*, *Ananas comosus*, *Kalanchoe fedtschenkoi*, and *Isoetes taiwanensis* [60]. In our experiments, leaves were taken at the same time of the day for the measurements of the level of gene expression. The expression level of almost all CA genes in adult, 2-month-old plants, after 16 days of acclimation to the changed CO_2_ concentrations in the air did not differ much from that in nCO_2_ plants of the same age (Figure 5 and Figure 6). The increase in the level of the *βca1.3*+*1.4* (Figure 5C)*, βca2* (Figure 5B), *βca3* (Figure 5D), and *βca4* (Figure 5A) transcripts encoding cytoplasmic CAs as well as of the *βca1.1*+*1.2* (Figure 6A) and *βca5* (Figure 6E) transcripts encoding chloroplast CAs in a short-term range 1–2 days after the beginning of the acclimation was preceded to the high CA activity (Figure 4). This result is surprising due to the convincing data of the role of βCA4 and βCA1 in CO_2_ transport into cells and in CO_2_-dependent regulation of stomatal permeability [14,15].

βCA5, presumably located in the thylakoid lumen [8,9], was the only thylakoid CA of which the expression level responded to the CO_2_ content in the air. The content of transcripts of the genes encoding thylakoid CAs, *αca2* and *αca4*, was not increased in lCO_2_ plants versus nCO_2_ ones, whereas the CA activities of both preparations of thylakoids, Thyl.1.0 and Thyl.0.3, were the highest in lCO_2_ plants (Figure 4D,E).

From two stromal Cas, only *βca1.1*+*1.2* transcripts showed an increase from exposure to lCO_2_ (Figure 6A). Thus, with a decrease in the concentration of carbon dioxide, it was intensified the synthesis of βCA1, i.e., of that CA, which participation in photosynthesis is constantly being questioned. The level of *αca1* transcripts was independent of CO_2_ content in the air (Figure 6B). However, In plants with knocked-out genes encoding αCA1, the number of indicators of photosynthetic activity, as well as the ability to accumulate starch, was decreased [61]. These data indicate that both stromal CAs, βCA1 and αCA1, as well as the thylakoid αCA5 (Figure 4D), play an important role in maintaining the Ci concentration close to carboxylation sites of Rubisco.

A significant decrease in the CA activity of the preparations of the chloroplast stroma and Thyl.1.0 in hCO_2_ plants versus nCO_2_ plants had no parallelism with the expression intensity of the corresponding genes. The expression levels of the genes encoding the stromal and thylakoid CAs in hCO_2_ plants were about the same or even slightly higher, against nCO_2_ plants (Figure 6). These data mean that the changes in the intensities of the synthesis of these CAs at the stage of transcription of the genes encoding them is not the main way of CA activity regulation, at least in the growth conditions used and at age of about two months. Apparently, this mechanism is associated, first of all, with the regulation of CAs’ activities, and to a lesser extent, with an increase in their biosynthesis at the level of the transcription of the genes encoding them. The mechanism of the regulation of CA activity was revealed for βCA1 by studying the action of high-temperature and water deficit stresses in the leaves of *Helianthus annum* [62] and *Brassica napus* [21]. The key regulation mechanisms of the CA activity in the chloroplasts in these plants were the nitration and phosphorylation of tyrosine residues in the active site of CA. The binding and dissociation of these nitrate and/or phosphate groups block and open, respectively, the passage of the substrate to the active site cavity according to plant needs. It is very likely that similar mechanisms of the regulation of CA activity exist for other CAs.

The additional studies on the effect of a decrease in the CO_2_ content in the air on the expression levels of CAs genes in Arabidopsis plants younger than two months have shown that in 26-days-old seedlings, these conditions caused a significant intensification of the synthesis of CA gene transcripts (Figure 7A). The expression levels of most cytoplasmic and chloroplast CAs genes were higher in 26-days-old plants after about two weeks of exposure to lCO_2_ versus nCO_2_ plants (Figure 7A). In 50 days-old plants after 16 days of acclimation to lCO_2,_ an increase in the intensity of the transcription of CA genes was less pronounced (Figure 7B). Thus, the younger the plants exposed to low CO_2_, the more the effect of an increase in the expression level of the genes encoding cytoplasmic and chloroplast CAs in them. Apparently, in young Arabidopsis plants, the increased intensity of the synthesis of these CAs at the stage of transcription of the genes encoding them makes a greater contribution to the increase in the content of CAs in the plant cell than that in mature plants.

Studies conducted using CA mutants, especially those with single mutations, most often demonstrate little pronounced effects on photosynthesis [15,22,63], except mutation in the *βca5* gene, which leads to significant suppression of the growth of Arabidopsis plants [36]. In recent years, more data on the involvement of CAs in higher plant cells in protection against stress or in transmitting a response signal to a negative environmental impact have begun to appear. Initially, these data were obtained for only βCA1 [34,35,63]. The functioning of βCA1 in these processes was ascribed to the possibility of βCA1 participation in fatty acids (FA) biosynthesis [64] and/or to βCA1’ss ability to bind salicylic acid (SA) [34]. These two assumptions, in fact, not only do not exclude, but also complement each other, since SAs and FAs are the key molecules of the stress-induced regulation of metabolic pathways. Recently, Hines et al. [37] have found that leaves of Δβ-ca1ca5 tobacco double mutants developed abnormally, and their leaves were significantly damaged from necrosis even when supplied with sucrose. Apparently, all six βCAs participated in these processes. Medina-Puche et al. [36] have shown the association of βCAs 1–6 from Arabidopsis with NPR receptors, which are the main participants of SA-induced stress signals. These authors concluded that βCAs are not involved in photosynthetic processes in higher plants. Wherein, our data show that under such stress as a low CO_2_ concentration in the air, a significant increase in CA activity occurs (Figure 4), but the intensity of *npr1* gene transcription, on the contrary, decreases (Figure 2C). These data show that at least under these conditions, an increase in the activity of CAs is not associated with their participation in the stress signal transmission through binding with the NPR1 protein.

In fact, the involvement of CAs in stress signaling does not exclude the possibility of their participation in photosynthetic processes. Dabrowska-Bronk et al. [38] have shown that in Arabidopsis plants all six βCAs are involved in the uptake of HCO_3_^−^ ions by roots, and their functioning is important for plant growth and cell homeostasis, especially under such stresses as lack of water and high light. The possible reason for the absence of dramatic effects of CAs mutations is that the CAs function together in plant cells, replacing each other in case of the suppression of the synthesis of any of them. In support of this hypothesis, it has been demonstrated that the addition of ethoxyzolamide, which is able to penetrate cell membranes and inhibit thus all cellular Cas, has led to a decrease in photosynthesis in the leaves of C3 and C4 plants at low CO_2_ concentrations [65] and at the CO_2_-dependent O_2_ release by pea leaf protoplasts both at low and optimal CO_2_ concentrations [66,67].

The described changes in the intensities of CAs gene expression, depending on the time of exposure to changed CO_2_ in the air and on the age of plants (Figure 5, Figure 6 and Figure 7), as well as the data of Hu et al. [15], DiMario et al. [2], Dabrowska-Bronk et al. [38], Medina-Puche et al. [36], and Hines et al. [37] indicate that the functioning of Cas in plants cells is carried out together, interdependently, and complexly. Our data show that this functioning depends significantly on the carbon dioxide content in the air, and this dependence is appreciably determined by the age of the plants.

## 4. Materials and Methods

### 4.1. Plant Material

Experiments were performed with Arabidopsis thaliana (L.) Heynh. ecotype Columbia-0 (Col). Three-week-old seedlings were planted into pots, one per pot, containing a commercially available soil mixture, and were grown in a growing chamber (CO_2_ content of 450 ppm, temperature 19 °C, 8 h day/16 h night photoperiod, photosynthetically active radiation of 50–70 µmol quanta m^−2^ s^−1^) for 25 days and then exposed to conditions of the low and high CO_2_ content, 150 and 1200 ppm under the same other conditions (Appendix A).

For the additional determination of the effects of age on the changes in the genes’ expression level after adaptation to low CO_2_, young Arabidopsis plants were grown in the conditions described above. At the age of 10 days (Appendix A) and 34 days (Appendix A) the plants were exposed to conditions of the low CO_2_ content (150 ppm). Redundant seedlings were removed from the pots of plants aged 10 days.

### 4.2. Measurement of Chlorophyll a Fluorescence

The maximum quantum yield of PSII (Fv/Fm), performance indexes (PI_abs_ and PI_total_), and the dissipation of light energy per active reaction center (DI_0_/RC) were calculated according to Kalaji et al. [68] after the measurement of OJIP chlorophyll *a* fluorescence kinetics. The OJIP chlorophyll *a* fluorescence transient was measured using a HandyPEA (Hansatech) fluorometer, with the leaves illuminated with a 1 s flash of red light of 3000 µmol quanta m^−2^ s^−1^. Before measurements, the plants were adapted to dark conditions for two hours.

### 4.3. Measurement of CO_2_ Assimilation Rate

The CO_2_ assimilation rate was measured in a leaf chamber using the LI-6800 Portable Photosynthesis System (Li-Cor, Lincoln, NE, USA) according to LI-6800 manual in the range of CO_2_ concentration of 0–1200 ppm under a constant light intensity of 350 µmol quanta m^−2^ s^−1^ (90% red, 10% blue light), 23 °C, and 50% relative humidity. Before measurement, the plants were pre-adapted to the illumination of 350 µmol quanta m^−2^ s^−1^ for 2 h. The leaf areas were measured using the Petiole application (Petiole LTD).

### 4.4. Determination of Starch and Soluble Carbohydrates Content

The starch content was analyzed by measuring the absorbance at 620 nm of leaf aqueous extracts supplemented with KI after thorough washing from pigments [69]. Prior measurements of leaves were kept on a wet filter paper for an hour in order to normalize a turgor of leaves. The content of soluble carbohydrates was determined in hydroalcoholic extract using phenol-sulfuric acid reaction according to Du Bois et al. [70].

### 4.5. Isolation of Cell Preparations

The isolation of the preparations of the cytoplasm, mitochondria, stroma, and thylakoids from Arabidopsis leaves was performed according to [7,41,42,71] with modifications (Appendix A). The leaves were homogenized in the Medium 1 containing 50 mM MES-Tris buffer (pH 8.2), 0.3 M sucrose, 40 mM NaF, 5 mM MgSO_4_, 1.5% Polyclar AT (*w/v*), 5 mM EDTA, 0.5% bovine serum albumin (BSA), 1 mM dithiothreitol (DTT), 1 mM benzamidine, 1 mM α-aminocaproic acid, and 1 mM phenylmethylsulfonyl fluoride (PMSF). The homogenate filtered through nylon cloth was centrifuged for 1.5 min at 150× *g* for the sedimentation of large fragments of leaves. The supernatant was centrifuged for 5 min at 2500× *g*, yielding the precipitate of chloroplasts and the supernatant “a”.

The supernatant “a” was centrifugated for 10 min at 8000× *g*, yielding precipitate (preparations of mitochondria) and supernatant “a′ ”, enriched with the proteins of cytoplasm. The preparations of the mitochondria obtained were suspended in Medium 1 and were used for analysis after incubation for 20 min with 1% Triton X-100.

The precipitate of the chloroplasts was suspended in Medium 2′ (Medium 2 diluted to one-tenth) to break the chloroplast envelope. Medium 2 contained 0.4 M sucrose, 35 mM K_2_HPO_4_, 15 mM NaH_2_PO_4_, 3 mM MgSO_4_**,** 10 mM KCl, 20 mM sodium ascorbate, 1 mM KHCO_3_, and 0.5 mM EDTA-Na, 1 mM DTT, 1 mM benzamidine, 1 mM α-aminocaproic acid, and 1 mM PMSF. The mixture was centrifuged for 5 min at 2500× *g*, yielding a precipitate of thylakoids and the supernatant “b” enriched with the proteins of chloroplast stroma.

Supernatants “a’ ” and “b” were additionally centrifuged for 1 h at 175,000× *g* to remove the rest of the membranes yielding preparations of cytoplasm and stroma, respectively, which were used for analysis. The preparations of the stroma were enriched with Rubisco and the preparations of the thylakoids and mitochondria had no Rubisco that was checked by a Western blot assay, using antibodies against the large subunit of Rubisco (Agrisera) (not shown).

The thylakoids were washed three times by suspending the pellets in Medium 2 followed by centrifugation for 5 min at 2500× *g*. Part of the preparations of the thylakoids was used for analysis after incubation for 20 min with Triton X-100 at a Triton/Chl ratio of 0.3 (Thyl.0.3). The other part of the preparations of the thylakoids was used for analysis after incubation for 20 min with Triton X-100 at a Triton/Chl ratio of 1.0 (Thyl.1.0).

### 4.6. Determination of the Protein Content

The protein content in the supernatants was determined using a DC^TM^ Protein Assay kit Bio-Rad according to Bio-Rad protocol.

### 4.7. Determination of the Chlorophyll Content

The chlorophyll content was determined in ethanol extracts according to Lichtenthaler [72].

### 4.8. Measurement of Carbonic Anhydrase Activity

Carbonic anhydrase activity was evaluated according to Khristin et al. [73] as the difference between the rates of pH decrease, measured with a pH electrode, from 8.3 to 7.8 in the course of CO_2_ hydration at 2 °C in 13.6 mM Veronal buffer (pH 8.4) in the presence and in the absence of an aliquot of the preparation. The difference in the buffer capacities was taken into account to express the CA activity as the extent of the proton release. The CA activity was calculated as the difference between the rates of the pH decrease in the presence and the absence of the preparation and was expressed in μmol H+ per 1 mg of Chl or protein per 1 min.

### 4.9. Quantitative Reverse Transcription PCR

Leaves were taken at the same time of day, namely, at 11 a.m. Total RNA was extracted from frozen Arabidopsis leaves, using the Aurum total RNA Mini Kit (BioRad), and treated with DNase to eliminate any genomic DNA contamination. Complementary DNA synthesis was performed using a reverse transcription kit OT-1 (Sintol) with oligo (dT15) as a primer. A quantitative reverse transcription polymerase chain reaction (qRT-PCR) was performed with qPCRmix-HS SYBR (Evrogen) and the primer pairs specific for genes coding COR414-TM1 (*At1g29395*) (forward 5′-GATAACCTAAGCGGATTGAAGCA-3′ and reverse 5′-ATCTTTCCACCACTGTGACTAAATCTAAACA-3′), ANAC019 (*At1g52890*) (forward 5′-CATAGAACCCAATCATCCAACTTAFTGCT-3′ and reverse 5′-AAAATAATCTCGACGGAAGGACAAAG-3′), LOX3 (*At1g74710*) (forward 5′-TCCAAGCGTGTGCTTACACCTC-3′ and reverse 5′-GTCCGTAACCAGTGATTGACAAG-3′), AOS (*At5g42650*) (forward 5′-GAGATTCGTCGGAGAAGAAGGAGAGAA-3′ and reverse 5′-AATCACAAACAACCTCGCCACCAAAA-3′, ICSI (*At1g74710*) (forward 5′-CAGCAGAAGAAGCAAGGCTT-3′ and reverse 5′-TCAATGCCCCAAGACCCTTTT-3′), PAD (*At3g52430*) (forward 5′-AGACTGGCGGGCATTACTTG-3′ and reverse 5′-CATCCAACCACTCTTTTGCTTGCTCA-3′), NPR1 (*At1g64280*) (forward 5′-GGAGAAGACGACACTGCTGAGAAA-3′ and reverse 5′-CACCGACGACGATGAGAGAG-3′), *At3g01500.1 + At3g01500.2* (*βca1.1+1.2*), *At3g01500.3 + At3g01500.4* (*βca1.3+1.4*), *At5g14740* (*βca2*), *At1g23730* (*βca3*), *At1g70410* (*βca4*), *At4g33580* (*βca5*), *At1g58180* (*βca6*), *At3g52720* (*αca1*), *At3g52720* (*αca2*), *At4g20990* (*αca4*). Primers sequences for the genes encoding CAs were used according to Rudenko et al. (2017), with primers for *βca1.1+1.2* corresponding to *βca1a* primers, and primers for *βca1.3+1.4* corresponding to *βca1b* primers. qRT-PCR data were normalized against Actin 7 gene. PCR reactions were performed in a LightCycler 96 Instrument, Roche Diagnostics GmbH.

## 5. Conclusions

In summary, this study demonstrated that in photosynthesizing cells, the conditions that require an increase in the intensity of inorganic carbon entry into cells lead to an increase in the activity of CAs located in the cytoplasm and CAs located in the chloroplasts, both stromal and thylakoid ones. Changes in the intensity of the expression of the genes encoding these CAs depended on the age of the plants. The increase in the level of CAs gene expression in lCO_2_ plants versus hCO_2_ plants was most pronounced in young 26-days-old plants. In 50 days-old plants, it was less noticeable, and in two-month-old plants, after 16 days of acclimation to the changed CO_2_ concentrations in the air, it did not differ much from that in nCO_2_ plants of the same age. The increase in the CA activities of the preparations of the cytoplasm, stroma, and thylakoids was preceded by an increase in the content of the transcripts of *βca1.3+1.4*, *βca2*, *βca3*, and *βca4* encoding cytoplasmic forms of CAs as well as of *βca1.1+1.2* and *βca5*, encoding chloroplast forms of CAs in a short-term range of 1–2 days after the beginning of the acclimation. The CA activity of the preparations of the mitochondria as well as the expression level of the gene encoding mitochondrial βCA6 was independent of the content of the carbon dioxide level in the air. These data do not support the assumptions of the participation of mitochondrial CAs in Ci supply to chloroplasts under conditions of low CO_2_.

Taken together, our data imply that CAs located in the cytoplasm and CAs located in the chloroplasts, both in stroma and in thylakoids, cooperatively participate in inorganic carbon supply to carboxylation sites in higher plants (Figure 8).

## Figures and Tables

**Figure 1 plants-11-02113-f001:**
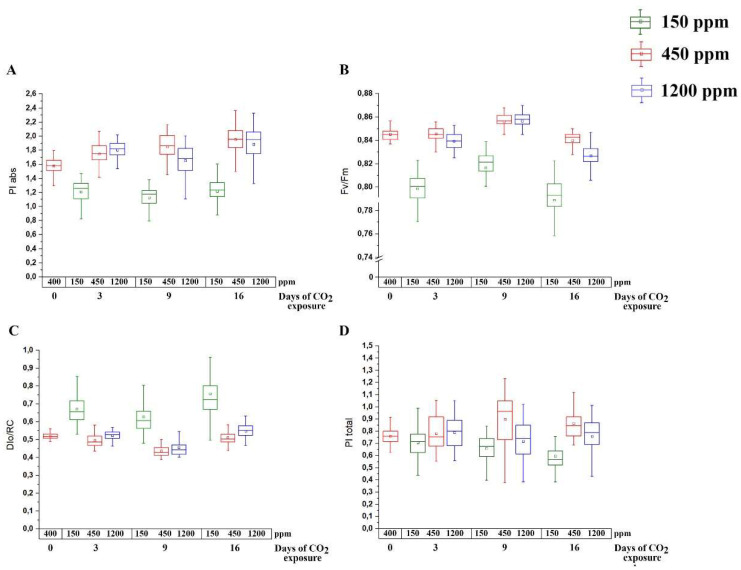
Effect of CO_2_ content in the air on photosynthetic parameters of Arabidopsis plants before (0), 3, 9, and 16 days after introduction to low (150 ppm), normal (450 ppm), and high (1200 ppm) CO_2_ levels in the air. Red boxes are control plants; green boxes are plants under low CO_2_ conditions; blue boxes are plants under high CO_2_ conditions. (**A**) PI_abs_, Photosynthetic performance index, which is estimated to absorption energy per photochemically active reaction center of PSII; (**B**) Fv/Fm, maximum quantum yield of PSII; (**C**) DI_0_/RC, dissipation of light energy per active reaction center; (**D**) PI_total_ performance index. Values are presented from three independent populations. The parameters of 30 leaves were measured in each population.

**Figure 2 plants-11-02113-f002:**
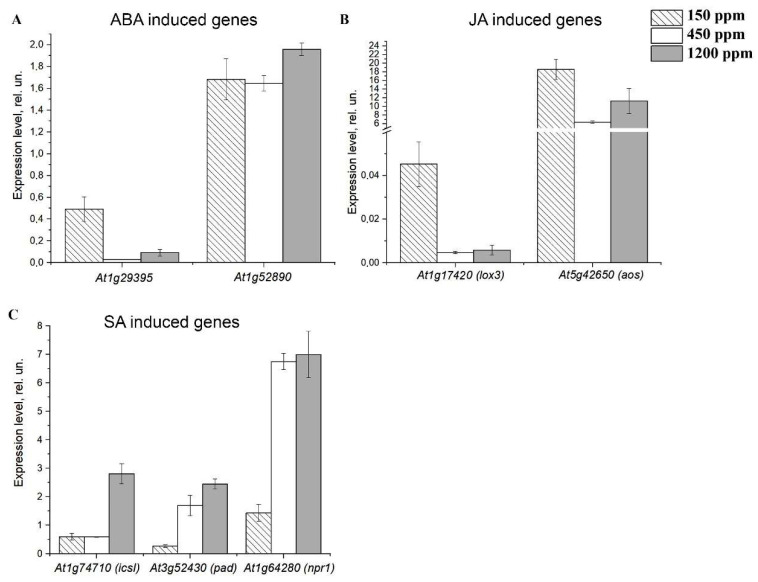
The content of transcripts of the genes inducible by plant immune signals in Arabidopsis plant leaves after 16 days of acclimation to low (150 ppm), normal (450 ppm), and high (1200 ppm) CO_2_ levels in the air. (**A**) *At1g29395* and *At1g52890* induced by abscisic acid; (**B**) *At1g17420* (*lox3*) and *At5g42650* (*aos*) induced by jasmonic acid; (**C**) *At1g74710* (*icsI*), *At3g52430* (*pad*) and *At1g64280* (*npr1*) induced by salicylic acid. Data were normalized for actin gene expression. Values are means ± S.E. of three independent experiments each formed by three repetitions.

**Figure 3 plants-11-02113-f003:**
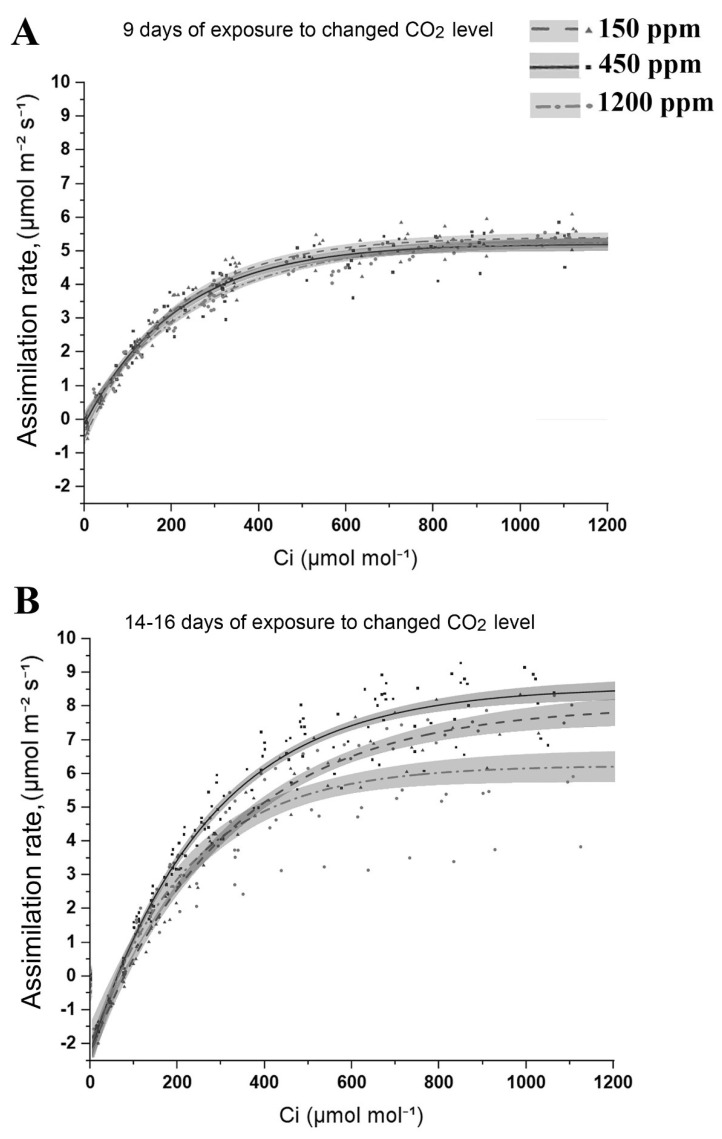
CO_2_ response curves measured using the dynamic assimilation technique at CO_2_ concentrations in the measurement chamber varying from 0 to 1200 ppm in leaves of Arabidopsis plants after 9 (**A**) and 14–16 days (**B**) of exposure to low (150 ppm) and high (1200 ppm) CO_2_ level. Assimilation rate in plants of the same age grown in the air with normal (450 ppm) CO_2_ level is the control. Curves show the data of a typical experiment.

**Figure 4 plants-11-02113-f004:**
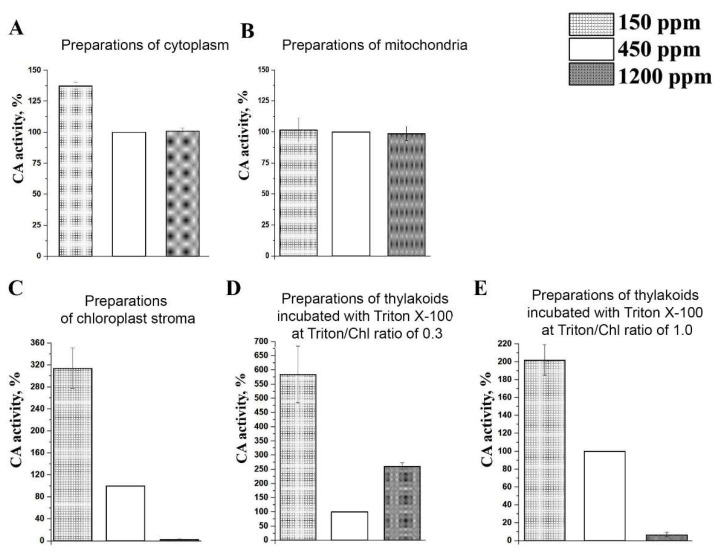
Carbonic anhydrase activities of preparations isolated as described in Materials and Methods from two-month-old Arabidopsis plants after 16 days of acclimation to low (150 ppm), and high (1200 ppm) CO_2_ levels in the air. The 100% value is the CA activity of preparations isolated from the leaves of Control plants of the same age grown at 450 ppm of CO_2_. Data are shown as mean ± the SE. (**A**)—preparations of cytoplasm. Briefly, 100% is 941.30 µmol H^+^ (min^−1^) (mg Pr^−1^). (**B**)—preparations of mitochondria, incubated with 1% Triton X-100. In total, 100% is 7093.50 (min^−1^) (mg Pr^−1^). (**C**)—preparations of chloroplast stroma. In addition, 100% is 9896.50 µmol H^+^ (min^−1^) (mg Pr^−1^); (**D**)—preparations of thylakoids incubated with Triton X-100 at Triton/Chl ratio of 0.3. In total, 100% is 148.5 µmol H^+^ (min^−1^) (mg Chl^−1^). (**E**)—Preparations of thylakoids incubated with Triton X-100 at a Triton/Chl ratio of 1.0. In brief, 100% is 201.00 µmol H^+^ (min^−1^) (mg Chl^−1^).

**Figure 5 plants-11-02113-f005:**
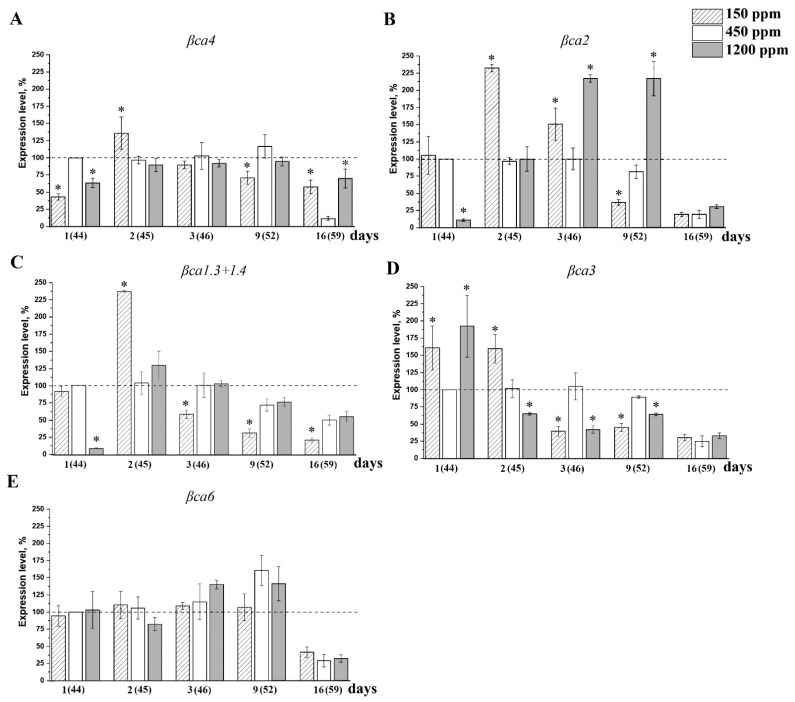
Levels of expression of genes encoding extrachloroplast carbonic anhydrases in Arabidopsis plant leaves after introduction to low (150 ppm), and high (1200 ppm) CO_2_ levels in the air. Numbers on the X axis show the age of the plants and (in parentheses) the number of days after moving introduction of the plants of 43 days of age grown at the atmospheric CO_2_ level (450 ppm) to conditions of changed CO_2_ level in the air. The 100% value (dashed horizontal line) is the expression level of the corresponding genes encoding CAs in the plants of 44 days of age grown at 450 ppm (Table 2). Data are shown as mean ± the SE. The experiments were performed three times with similar results. (**A**)—*βca4*; (**B**)—*βca2*; (**C**)—*βca1.3+1.4*; (**D**)—*βca3*; (**E**)—*βca6*. Asterisks denote statistically significant differences between values for different CO_2_ levels on the same experimental day, *p* < 0.01.

**Figure 6 plants-11-02113-f006:**
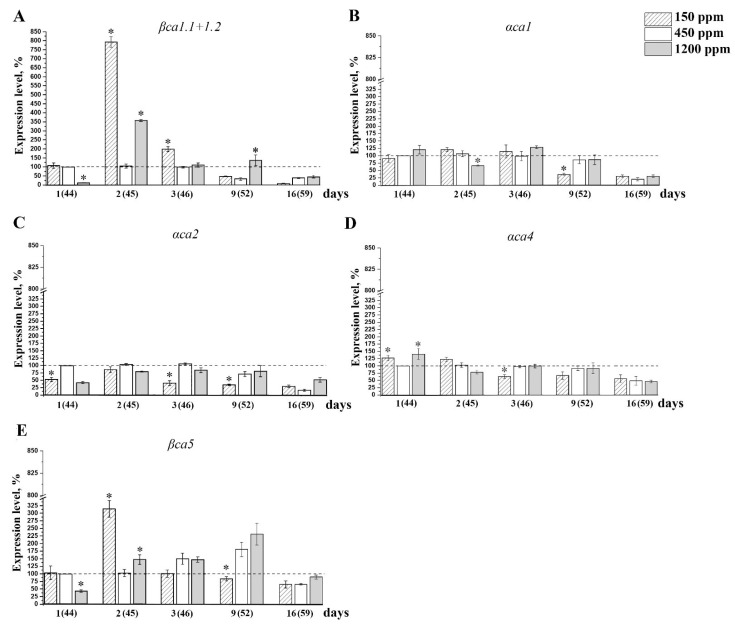
Levels of expression of genes encoding chloroplast carbonic anhydrases in Arabidopsis plant leaves after introduction to low (150 ppm), and high (1200 ppm) CO_2_ levels in the air. Numbers on the *X*-axis show the number of days after moving the plants of 43 days of age grown at 450 ppm at the atmospheric CO_2_ level to conditions of changed CO_2_ level in the air. The 100% value (dashed horizontal line) is the expression level of the corresponding genes encoding CAs in the plants of 44 days of age grown at 450 ppm (Table 2). Data are shown as mean ± the SE. The experiments were performed three times with similar results. (**A**)—*βca1.1*+*1.2*; (**B**)—*αca1*; (**C**)—*αca2*; (**D**)—*αca4*; (**E**)—*βca5*. Asterisks denote statistically significant differences between values for different CO_2_ levels on the same experimental day, *p* < 0.01.

**Figure 7 plants-11-02113-f007:**
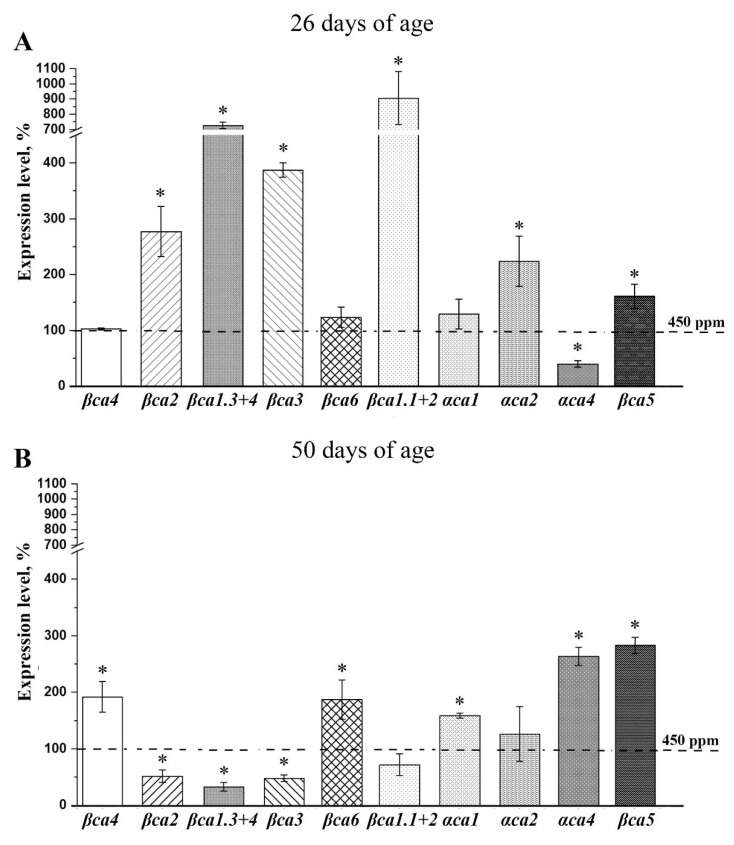
Levels of expression of genes encoding CAs in leaves of Arabidopsis plants of 26 (**A**) and 50 (**B**) days of age after 16 days introduction to low (150 ppm) CO_2_ level in the air. The 100% value (dashed horizontal line) is the expression level of the corresponding genes encoding CAs in the Control plants, according to Table 2, for 26 (**A**) and for 50 (**B**) days of age grown at 450 ppm. Data are shown as mean ± the SE. Asterisks denote statistically significant differences between values for different CO_2_ levels on the same experimental day, *p* < 0.01.

**Figure 8 plants-11-02113-f008:**
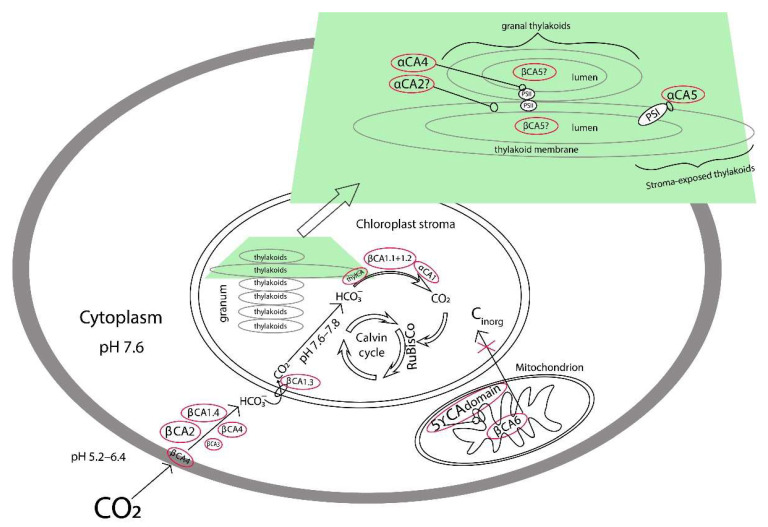
The hypothetical scheme of CAs positions and cooperative participation of CAs located in the cytoplasm and CAs located in the chloroplasts in inorganic carbon supply to carboxylation sites in higher plants. Mitochondrial CAs, apparently, are not involved in inorganic carbon supply to chloroplasts.

**Table 1 plants-11-02113-t001:** The content of starch and soluble carbohydrates in Arabidopsis plant leaves after 16 days of acclimation to low (150 ppm), normal (450 ppm), and high (1200 ppm) CO_2_ levels in the air. The table shows the data of a typical experiment. In the parentheses are given the data in % of those from control values, i.e., from data for plants grown at 450 ppm CO_2_.

CO_2_ Level in Air	Starch Content, mg/g of Fresh Weight	Soluble Carbohydrates Content, mg/g of Fresh Weight
150 ppm	0.82 ± 0.04 (23%)	2.92 ± 0.35 (74%)
450 ppm	3.60 ± 0.34 (100%)	3.95 ± 0.31 (100%)
1200 ppm	2.19 ± 0.22 (61%)	4.25 ± 0.36 (108%)

**Table 2 plants-11-02113-t002:** Levels of expression of genes encoding CAs in Arabidopsis plants leaves of different ages grown at the CO_2_ content of 450 ppm, temperature 19 °C, short day photoperiod 8 h day/16 h night, photosynthetically active radiation of 50–70 µmol quanta m^−2^ s^−1^. Data were normalized for actin gene expression. Data are shown as mean ± the SE of three independent experiments each formed by four repetitions.

Genes	Levels of Expression at			
	26^th^ Day of Age	44^th^ Day of Age	50^th^ Day of Age	59–60^th^ Days of Age
*βca4*	1037.9 ± 12.46	60.6 ± 2.7	55.89 ± 3.97	7.22 ± 0.52
*βca2*	1416.38 ± 12.35	195.67 ± 9.41	190.88 ± 15.41	37.90 ± 6.92
*βca1.3+1.4*	1290.03 ± 7.46	153.08 ± 16.1	140.74 ± 12.30	76.54 ± 5.43
*βca3*	1.52 ± 0.20	0.98 ± 0.04	0.80 ± 0.02	0.25 ± 0.07
*βca6*	10.94 ± 0.09	7.99 ± 0.93	6.45 ± 0.32	2.94 ± 0.06
*βca1.1+1.2*	24.34 ± 4.67	19.96 ± 0.96	17.91 ± 0.55	7.91 ± 0.46
*αca1*	69.48 ± 5.71	16.43 ± 0.78	14.04 ± 1.44	3.31 ± 0.27
*αca2*	3.8 ± 0.28	0.93 ± 0.05	0.71 ± 0.09	0.16 ± 0.05
*αca4*	0.78 ± 0.03	0.28 ± 0.01	0.26 ± 0.04	0.14 ± 0.02
*βca5*	14.8 ± 2.3	10.59 ± 0.13	10.3 ± 0.20	6.98 ± 0.25

## Data Availability

Not applicable.

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
