# Peer review of "Effect of CO2 Content in Air on the Activity of Carbonic Anhydrases in Cytoplasm, Chloroplasts, and Mitochondria and the Expression Level of Carbonic Anhydrase Genes of the α- and β-Families in Arabidopsis thaliana Leaves"

_plants, 2022, doi:10.3390/plants11162113_

Round 1
Reviewer 1 Report
Manuscript ID: plants-1797641
Title: Effect of CO2 Content in Air on the Activity of Carbonic Anhydrases in Cytoplasm, Chloroplasts and Mitochondria and the Expression Level of Carbonic Anhydrase Genes of the α- and β-Families in Arabidopsis thaliana Leaves
Authors: Natalia N. Rudenko, Lyudmila K. Ignatova, Ilya A. Naydov, Natalia S. Novichkova, Boris N. Ivanov
Recommendation: Major revision
The authors present interesting and complex work on carbonic anhydrase. The Manuscript is well written, but sometimes is not easy to follow, especially Discussion. Some parts of the MS are too detailed. Also, Discussion section is too long, could be more concise. Some of methods used are not well described, only reference was written or none at all. You can find my comments below:
Line 16 - Absract – remove OJIP and kinetics from the sentence. Only “Chlorophyll fluorescence” is absolutely enough in this case, in my opinion.
Fig. 1, 2, 3, 4, 6 - Please, add more information on the graphs about which panel which investigated fraction is (Figs. 4,6), what means green, red, blue colors (Fig. 1), days (Fig. 3), SA-, JA- or ABA induced (Fig. 2). It will be easier for the reader. Using the name for example At1g64280 in Results is a little confusing. After, in Discussion, the authors use npr1.
Fig. 7 - Figure legend is totally missing.
Line 137 – The same like in line 16, “fast chlorophyll a fluorescence curves (OJIP)” maybe is better to be changed to “chlorophyll a fluorescence”. From the parameters presented is clear that the authors are using PEA to measuring chl fluorescence, but OJIP curves are not presented.
Lines 152-155, 192-194, 200-203, 222-225, 239-242, 283-287, 324-327, 360-365 – The authors discuss the obtained results in Results section. These parts should be moved to Discussion.
Line 398 - Replace intensity with level
Line 403 – “The growth of carbon dioxide in the air…” – Revise the sentence, this part has no mean, something is missing
Line 609 - M&M 4.3. Measurement of CO2 assimilation rate – Why you preadapted the plants to 350 micromol PAR for measuring the CO2 response curves, when they are growing at 50-70 micromol? Is it possible that the plants exhibit photoinhibition because of the high light used for preadaptation? Maybe the used light influenced the obtained results for CO2 assimilation rate (lowering the values).
Author Response
Recommendation: Major revision
The authors present interesting and complex work on carbonic anhydrase. The Manuscript is well written, but sometimes is not easy to follow, especially Discussion. Some parts of the MS are too detailed. Also, Discussion section is too long, could be more concise. Some of methods used are not well described, only reference was written or none at all. You can find my comments below:
Response: Thank you for your professional comments. We corrected Discussion and shortened the list of references according to the corrections made. We added the clarifications to the Methods section in the new manuscript. Thank you!
Line 16 - Absract – remove OJIP and kinetics from the sentence. Only “Chlorophyll fluorescence” is absolutely enough in this case, in my opinion.
Response: we corrected it in the new manuscript. Thank you!
Fig. 1, 2, 3, 4, 6 - Please, add more information on the graphs about which panel which investigated fraction is (Figs. 4,6), what means green, red, blue colors (Fig. 1), days (Fig. 3), SA-, JA- or ABA induced (Fig. 2). It will be easier for the reader.
Response: we added this information to the Figures for the new manuscript. Thank you!
Using the name for example At1g64280 in Results is a little confusing. After, in Discussion, the authors use npr1.
Response: we corrected it in the new manuscript. Thank you!
Fig. 7 - Figure legend is totally missing.
Response: we added the legend to Figure 7 it in the new manuscript. Thank you!
Line 137 – The same like in line 16, “fast chlorophyll a fluorescence curves (OJIP)” maybe is better to be changed to “chlorophyll a fluorescence”. From the parameters presented is clear that the authors are using PEA to measuring chl fluorescence, but OJIP curves are not presented.
Response: we corrected it in the new manuscript. Thank you!
Lines 152-155, 192-194, 200-203, 222-225, 239-242, 283-287, 324-327, 360-365 – The authors discuss the obtained results in Results section. These parts should be moved to Discussion.
Response: Thank you for the comments. The conclusions from the lines 324-325 were moved to Discussion. The other specified conclusions, in the Results sections 2.1 and 2.2. are necessary for understanding the choosing of acclimation time. In addition, it was necessary, before presenting the following results, to explain that the plants are under stress, both under lCO2, as well as under hCO2 with different mechanisms of plant response to lCO2 and hCO2. The conclusions in the rest sections of Results are important for making the rather complex results of our research easier to understand up to the Discussion section.
Line 398 - Replace intensity with level
Response: In the modified Discussion section, this part is absent.
Line 403 – “The growth of carbon dioxide in the air…” – Revise the sentence, this part has no mean, something is missing
Response: In the modified Discussion section, this part is absent.
Line 609 - M&M 4.3. Measurement of CO2 assimilation rate – Why you preadapted the plants to 350 micromol PAR for measuring the CO2 response curves, when they are growing at 50-70 micromol? Is it possible that the plants exhibit photoinhibition because of the high light used for preadaptation? Maybe the used light influenced the obtained results for CO2 assimilation rate (lowering the values).
Response: Thank you for the comment.
The preliminary measurements of CO2 assimilation rate, which we carried out with the selection of conditions for the experiment, always resulted in a clipped curve of the CO2 response at 50-70 microE. Apparently, this was due to the assimilation was light-limited at high CO2 concentrations.
Without preadaptation, when the assimilation curve is recorded at 350 microE, the plants status changes during the recording process leading to a skewed CO2 assimilation curve (i.e. CO2 assimilation rate at 400 ppm at the beginning of the recording is different from that at the end of the recording). Therefore, we carried out the preadaptation to 350 microE for the amount of time needed for CO2 assimilation rate and stomatal conductance index to reach stable values at 400 ppm.

Reviewer 2 Report
Rudenko et al. present in their article "Effect of CO2 content in air on the activity of carbonic anhydrases in cytoplasm, chloroplasts and mitochondria and the expression level of carbonic anhydrase genes of the α- and β-families in Arabidopsis thaliana leaves" a very detailed investigation of the important carbonic anhydrase. Overall this is a very well-conducted study.
The authors present a whole array of data. So much, that I would suggest a summary figure that consolidates all garnered results. This will provide a much better overview and increase accessibility of the manuscript.
Figure 7 has no figure legend. This has to be amended.
Regarding the expression patterns of carbonic anhydrase, I would like the authors to highlight that they can also follow a daytime-dependent expression, as in the CAM plants A. comosus and K. fedtschenkoi and Isoetes, see and cite:
Wickell, D., Kuo, LY., Yang, HP. et al. Underwater CAM photosynthesis elucidated by Isoetes genome. Nat Commun 12, 6348 (2021). https://doi.org/10.1038/s41467-021-26644-7
Author Response
Rudenko et al. present in their article "Effect of CO2 content in air on the activity of carbonic anhydrases in cytoplasm, chloroplasts and mitochondria and the expression level of carbonic anhydrase genes of the α- and β-families in Arabidopsis thaliana leaves" a very detailed investigation of the important carbonic anhydrase. Overall this is a very well-conducted study.
The authors present a whole array of data. So much, that I would suggest a summary figure that consolidates all garnered results. This will provide a much better overview and increase accessibility of the manuscript.
Response: Thank you for the suggestion. We added the Graphical Abstract, which consolidates the modern knowledge of the positions of CAs in higher plants cells with our main results.
Figure 7 has no figure legend. This has to be amended.
Response: we added the legend to Figure 7 it in the new manuscript. Thank you!
Regarding the expression patterns of carbonic anhydrase, I would like the authors to highlight that they can also follow a daytime-dependent expression, as in the CAM plants A. comosus and K. fedtschenkoi and Isoetes, see and cite:
Wickell, D., Kuo, LY., Yang, HP. et al. Underwater CAM photosynthesis elucidated by Isoetes genome. Nat Commun 12, 6348 (2021). https://doi.org/10.1038/s41467-021-26644-7
Response: Thank you for the comment. We added the appropriate text to Discussion, Methods and References section.

Round 2
Reviewer 1 Report
Manuscript ID: plants-1797641
Title: Effect of CO2 Content in Air on the Activity of Carbonic Anhydrases in Cytoplasm, Chloroplasts and Mitochondria and the Expression Level of Carbonic Anhydrase Genes of the α- and β-Families in Arabidopsis thaliana Leaves
Authors: Natalia N. Rudenko, Lyudmila K. Ignatova, Ilya A. Naydov, Natalia S. Novichkova, Boris N. Ivanov
The authors have corrected the MS according to almost all of my suggestions.
My main concern:
- Results section contains more than 30 lines that in my opinion, belong to Discussion – the authors moved only 2 lines from Results to Discussion
From v.1 of the MS: “Lines 152-155, 192-194, 200-203, 222-225, 239-242, 283-287, 324-327, 360-365 – The authors discuss the obtained results in Results section. These parts should be moved to Discussion.
Response: Thank you for the comments. The conclusions from the lines 324-325 were moved to Discussion. The other specified conclusions, in the Results sections 2.1 and 2.2. are necessary for understanding the choosing of acclimation time. In addition, it was necessary, before presenting the following results, to explain that the plants are under stress, both under lCO2, as well as under hCO2 with different mechanisms of plant response to lCO2 and hCO2. The conclusions in the rest sections of Results are important for making the rather complex results of our research easier to understand up to the Discussion section.”
Other comments:
- All figures are presented in low quality
- Figure 1, 3, 4, 5 – in the legend correct CO2 to CO2, –1 – superscript
- Figures 1 and 2 are placed after Figure 7 in the MS, in Discussion section
- Different fonts and highlighted text could be seen in the MS
- At the end of the MS, tables with weird numbers appeared
Author Response
Title: Effect of CO2 Content in Air on the Activity of Carbonic Anhydrases in Cytoplasm, Chloroplasts and Mitochondria and the Expression Level of Carbonic Anhydrase Genes of the α- and β-Families in Arabidopsis thaliana Leaves
Authors: Natalia N. Rudenko, Lyudmila K. Ignatova, Ilya A. Naydov, Natalia S. Novichkova, Boris N. Ivanov
The authors have corrected the MS according to almost all of my suggestions.
My main concern:
- Results section contains more than 30 lines that in my opinion, belong to Discussion – the authors moved only 2 lines from Results to Discussion
From v.1 of the MS: “Lines 152-155, 192-194, 200-203, 222-225, 239-242, 283-287, 324-327, 360-365 – The authors discuss the obtained results in Results section. These parts should be moved to Discussion.
Response: Thank you for the comments. The conclusions from the lines 152-155, 192-194, 200-203 were moved from Results section to Discussion.
The conclusions from the lines 222-225 were emphasized in Discussion. This conclusion is important Results section, for better understanding the following results.
The conclusion from the lines 239-242 is important for understanding the sections 2.3 and 2.4. of Results.
The conclusions from the lines 283-287 were deleted. These conclusions are described in details in the lines 472-490 of Discussion.
The conclusions from the lines 324-325 have already been moved to Discussion.
The conclusion from the lines 360-365 is important for understanding the section 2.5. of Results. This is one of the key results of the present study. It was especially emphasized both in the Discussion and in the Conclusion.
Other comments:
- All figures are presented in low quality
Response: All of our Figures are in 350-400 dpi. To avoid a low quality of the Figures in the final file, we are sending our new Figures files as a .zip file
- Figure 1, 3, 4, 5 – in the legend correct CO2 to CO2, –1 – superscript
- Figures 1 and 2 are placed after Figure 7 in the MS, in Discussion section
- Different fonts and highlighted text could be seen in the MS
- At the end of the MS, tables with weird numbers appeared
Response: The described problems arose, apparently, during the layout of the text. To avoid these problems, we are sending our new file as pdf file.
